# Patient-Centric Medicine Design: Key Characteristics of Oral Solid Dosage Forms that Improve Adherence and Acceptance in Older People

**DOI:** 10.3390/pharmaceutics12100905

**Published:** 2020-09-23

**Authors:** Zakia Shariff, Daniel Kirby, Shahrzad Missaghi, Ali Rajabi-Siahboomi, Ian Maidment

**Affiliations:** 1Aston Pharmacy School, Aston University, Birmingham B4 7ET, UK; i.maidment@aston.ac.uk; 2Colorcon Inc., Harleysville, PA 19438, USA; SMissaghi@colorcon.com (S.M.); Asiahboomi@colorcon.com (A.R.-S.)

**Keywords:** patient centric, oral solid, acceptance, adherence, older people

## Abstract

Older people represent a very heterogeneous patient population and are the major user group of medication. Age-related changes mean that this population can encounter barriers towards taking medicines orally. The aim of this study was to investigate the characteristics of oral solid dosage forms that contribute to an age appropriate dosage design, with an aim to improve overall medication adherence and acceptance in older people. Fifty-two semistructured interviews were conducted with older people, informal (family) carers, and health and social care professionals. Formulation characteristics impacted three stages of the medication taking process: (1) medication identification and memorability, (2) medication handling and (3) swallowability. Small round tablets (≤7 mm) are least accepted amongst older people and their carers and had a negative impact on all stages. The use of bright, two-coloured preparations and interesting shapes improves identification and further aids memorability of indications and the timing of tablets. Palatability, while useful to enhance swallowability, also has an impact on the visual appeal and memorability of medication. Environmental, patient, medication and disease characteristics also determine preferences for formulation. Developing an age appropriate dosage design for older people, therefore, requires a holistic, patient-centric approach to improve adherence and acceptance.

## 1. Introduction

The term “patient centricity” is increasingly being used in the literature and has gained interest both within academia and the pharmaceutical industry. There is a growing trend for pharmaceutical companies to ensure that the end product takes into account the needs of the user to enhance acceptability; patient-centric medicine design is also an essential tool to improve the patients’ quality of life [1]. Patient-centric medicine design has been defined as the process of identifying and addressing the comprehensive needs of a target population, resulting in a formulation that provides the best overall benefit to risk profile [1]. The older population, the major user group of prescribed medicines, has been the focus of recent recommendations published by both the European Medicines Agency (EMA) and Food and Drugs Administration (FDA) [2,3]. These guidelines state the need to consider differences in physical characteristics (e.g., size and shape of the tablet or capsule) and how these may affect patient compliance and acceptability of medication regimens [3]. When discussing these characteristics, both the EMA and FDA refer to “patient acceptability” [3,4], defined as “an overall ability of the patient and caregiver to use a medicinal product as intended” [5]. Acceptability can have a significant impact on adherence, defined by NICE as “the extent to which the patient’s action matches the agreed recommendations” [6]. The design of a patient-centric drug product for the older population, however, presents a challenge due to the heterogeneous nature of this population [7]. Co-morbidities and age-related differences, including changes in cognition, motor functions and sensory functions, need to be considered. Changes in motor functions, for example, can lead to reduced hand–eye coordination, trembling hands, impaired manual dexterity and dysphagia [4]. This can have a significant impact on the medication taking process, which involves a number of steps, including identifying the correct medicine, reading and understanding information labels, handling and removing the outer packaging, preparing the drug before administration (if required) and taking the medicine [8]. Further challenges associated with drug therapy for older people include the increased roles and responsibilities of informal (family) carers within the community. This is especially true for conditions such as dementia, where the role of medication management is often passed to informal carers who can find this role challenging [9].

The lack of appropriate licensed dosage forms for this population results in dosage forms being routinely modified, with medications affecting the Central Nervous System being the most frequently modified [10]. While modifications are sometimes necessary to facilitate fractional dosing and to overcome swallowing difficulties, studies and guidelines recommend that modifications are best avoided due to the legal and clinical risks that may arise [11]. In particular, alterations can further complicate treatment by potentially changing the bioavailability, toxicity and stability of medicines, especially for those oral solid dosage forms with a functional coating and/or modified release properties (such as modified release preparations of co-careldopa, nifedipine and metformin) [12]. Formulation scientists must therefore work closely with both patients and informal carers in order to develop an age appropriate formulation that improves patient adherence [13]. There is also a need to include the views of health and social care professionals to provide the multidisciplinary approach that is needed for medication optimisation within this population [14]. However, a recent systematic review published in this area found only a single study that involved the views of General Practitioners and no studies that involved formal (paid) or informal (family) carers [15]. Further qualitative work is required to provide an insight into the issues experienced by older people and informal carers when using/administering oral solid dosage forms [15]. This study, therefore, aims to use a qualitative approach to investigate the characteristics of oral solid dosage forms that will contribute to an age appropriate dosage design, with the aim to improve overall medication adherence and acceptance in older people.

## 2. Materials and Methods

Purposeful sampling was used to identify older people (aged 65 or above taking at least one oral solid dosage form), their informal carers and health/social care professionals (including formal carers) with experience of working with older people. Older people and their carers unable to speak English or lacking capacity to consent were excluded. The study was advertised on recruitment websites (including Join Dementia Research and People in Research), within care homes (via ENRICH (Enabling Research in Care Homes) and RMBI (The Royal Masonic Benevolent Institution)) and also via primary and secondary care NHS organisations. All interviews took place at a mutually convenient location. For patients and informal carers, this often included the patient’s home as participants may not be as open in their answers in a strange environment [16].

### 2.1. Study Design and Data Collection

Semistructured interviews were used to collect qualitative data [17]. The interview schedules for older people, carers and health and social care practitioners were developed through informal discussions with patients and care staff, the principle investigator’s experience as a practising community pharmacist and through discussion with the research team. The interview schedules (Table 1) were used as a guide, and follow up questions were asked based on the responses of participants. The principle investigator conducted all interviews. The sample size was determined by data saturation and this has been defined as the point at which “no new information or themes are observed” [18]. This was, therefore, the key criterion for discontinuing data collection.

In addition to the interview schedule, participants were provided with placebo tablets (Figure 1) as a point of reference that helped them to communicate their ideas. Previous studies in adults have found that tablet sizes greater than approximately 8 mm in diameter are associated with more complaints [19]. Furthermore, oval tablets, especially larger ones, are easier to swallow than round ones in healthy volunteers (under the age of 65) [20]. In order to explore this in older people, placebo tablets of sizes both greater than and less than 8 mm were made available to participants, alongside oval, caplet and round shaped tablets. The placebo tablets were developed in partnership with Colorcon^®^ and were not tasted or swallowed by participants.

### 2.2. Qualitative Analysis

All Interviews were audio recorded and transcribed verbatim using an approved transcription service. Interview transcriptions were then rechecked for accuracy by the principle investigator.

Thematic analysis was used to analyse the data [21]. This involves six key stages: familiarisation with data, generation of initial codes, searching for themes, reviewing themes, defining and naming themes and finally producing the report [21]. All themes and codes were reviewed with members of the research team at each stage to reduce any potential for bias.

### 2.3. Ethics

The study received NHS HRA approval and approval from the Social Care REC (18/IEC08/0047, approved on 14 January 2019). All participants received written information about the study and gave written informed consent.

## 3. Results

A total of 52 interviews were conducted; 18 older people (with a mean age of 78.7 years and a range of 66 to 97 years), 7 informal carers (with a mean age of 60.9 years and a range of 48 to 70 years) and 27 health/social care professionals. The characteristics of included participants are shown in Appendix A. Three key themes were extracted from the data, all of which explored the impact of the medication’s characteristics on different stages of the medication taking process: (1) medication identification and memorability, (2) medication handling and (3) swallowability.

### 3.1. Medication Identification and Memorability

#### 3.1.1. Colour

The usefulness of colour to aid visual identification of tablets was dependent on setting; some older people within care homes had fewer preferences for colour and described the colour as “incidental”. In contrast, older people living alone within the community described the importance of using bright colours to ensure tablets are easily visible, especially when they were accidentally dropped on the floor.
“I think colour and I think bright colour. I would say bright colour. If they’re wishy-washy pale pink, pale yellow and white, they’re all a bit similar. I think colour would be good and reasonably bright colour.”(P5)

Brighter colours were described as being particularly important for people living with dementia, who require “visible” and “appealing” colours. Healthcare professionals also referred to the importance of brighter colours due to a decline in visual acuity as a result of macular degeneration or cataracts:
“Is there something that maybe needs to be brighter because they’ve got, maybe, macular degeneration or they’ve got problems with cataracts.”(HCP11)

Colour was also a useful tool to differentiate between medications, especially for patients taking a large number of tablets. Older people and informal carers often described the difficulties encountered when tablets were all white and this was highlighted as a deterrent to using pill organisers. Participants suggested the use of the same colour for groups of medications that had similar indications to help identification. A patient living with dementia further suggested the use of colour to differentiate between the timing of different medications. Using colour to explain and differentiate between medications was described by informal carers as being particularly important for older people who do not understand English. Healthcare professionals agreed, suggesting overall understanding and, therefore, adherence could be improved if colour could be used for groups of medications when counselling patients.
“If there are a few tablets that are really important, for example anticoagulants … if they have a specific colour maybe it’s easier even, even for the clinician to say “you know the brown tablet or the red tablet, you need to take.” (HCP25)

Colour was also seen as helpful to improve memorability of medications; healthcare professionals, older people and carers all referred to the use of colour as a tool to remember which medications were being taken. Distinctive colours were useful to remind patients whether they had adhered to a certain tablet and this was especially important due to the number of tablets being taken. Older people and informal carers further suggested that, based on previous experience, preparations with two colours were especially easy to remember:
“Now, these particular ones, without opening it, I think are only white, though I have had some that are white and red, and that, actually, would be a lot better because then I would remember better, I think, when it comes to relating what I’ve done with the Paracetamol … if there was a query in my head I might be able to remember better if it had more of a distinctive combination.”(P3)

#### 3.1.2. Dimensions and Markings

The size of tablets was also important to aid visual identification. Smaller tablets (similar to the 6 mm round) were described as difficult to see and often led to unintentional non-adherence. This was a significant concern for informal carers, who reported concern over whether smaller tablets (such as folic acid and Glimepiride) could be seen by the person they care for. The very slight discrepancy between smaller sizes, such as the 6 mm and 7 mm round tablets, was also described as having the potential to cause confusion, and medication errors were reported as a result of difficulties differentiating between tablets. Informal carers took on the responsibility of weighing up the importance of the tablet and discussed the need to ask for an alternative from the doctor if “more important” medications were of a small size to ensure patient adherence. There was a general preference towards larger tablets amongst this group to help ease the medication administration process; however, this was balanced alongside the ease of swallowing for the patient (Theme 3):
“The larger they are, the easier. I mean I’m not sure about how that would be for my dad, he doesn’t seem to have any problem swallowing them but in terms of, you know, seeing them and making sure that they’re there in his hand” (C5)

Shape was also an important tool to differentiate between medications and aid memorability. Unusual shapes were easier to remember and could be associated with the name of the drug. Participants further referred to the potential to use shape to associate a tablet with the time of day it needed to be taken; for example, the use of a star shape for tablets to be taken at night. While not unusual, the common use of an oval shape for statins led to participants associating these together and patients used phrases such as “the little oval one that I have at night time” when describing these tablets to health care professionals.
“… I always recognise the Amlodipine… because it’s sort of got one, two, three, four, about six or eight sided it is. It’s quite small but it’s an interesting shape, so I do notice that” (P5)

When shape, colour and size are all similar, some participants suggested the use of markings to help differentiate between medications. Informal carers, in particular, highlighted this would improve acceptance, as it would reduce the worry when administering two tablets which look similar. There was an emphasis on the need for these markings to relate to the name or strength of the drug, and healthcare professionals further discussed the need for these to be easily visible:
“I am just thinking, for instance, sort of like a Statin and a Doxazosin, the imprint on the tablet’s not very defined. And that’s where some mistakes have actually happened.” (HCP12)

#### 3.1.3. Impact of Changes in Appearance on Identification and Memorability

The appearance of oral solid dosage forms is an important identification tool; however, challenges can arise when different brands of the same medication are dispensed. These changes can be especially difficult for patients with conditions such as Alzheimer’s disease, who rely on the appearance of tablets. Dispensing new brands led to the need for some care professionals to actively request a reversion to the old brand at the pharmacy due to non-adherence. Changes in appearance, in general, were not explained to patients or informal carers by healthcare professionals, and older people described the anxiety that this often led to, with some fearing that they had received somebody else’s medication by mistake.
“They’re quite difficult sometimes because when I get my medication, it’s obviously generic so a different manufacturer will give a different colour, or no colour at all for that matter or a different size and you know if you’re taking, like I say, Memantine, if it changes colour you think, well I wonder what that is I’ve got there.”(P2)

### 3.2. Medication Handling

#### 3.2.1. Difficulties Removing and Handling

Small round tablets (≤6 mm round) were difficult to remove from blisters, dosettes and weekly pill organisers. Informal carers described the need to double check that all medications were taken, due to the likelihood of unintentional non-adherence, and reported concern in relation to how the patient would take their medication if they were not available to administer it.
“I mean it’d make it easier for her if she ever needed … couldn’t have a blister pack, and there wasn’t somebody to administer them, if they’re easier to get out of the packaging, because some aren’t”(C3)

In order to overcome these challenges, some healthcare professionals suggested the possibility of assessing an older person’s ability to remove medication from the packaging prior to dispensing using samples of tablets. Participants referred in particular to rheumatoid arthritis, stroke, neuropathy and carpal tunnel syndrome, all of which affected older peoples’ ability to remove and handle smaller tablets.
“It’s about asking about them … do you feel that you can take it, can you pick it up, and it might be even asking, you know, samples, so can you show me whether or not you can pop it out of the blister pack or out of the packaging.”(HCP11)

Following removal, difficulties were also highlighted by older people, carers and healthcare professionals when handling small, round tablets prior to administration. Both the 6 mm and 7 mm round tablets were least preferred and healthcare professionals referred to the potential for these to be dropped on the floor, resulting in distress for the patient. Informal carers within the community often took on the responsibility of finding tablets that had been dropped; however, this was dependent on the appearance of the tablet and how easily visible it was (Theme 1). Older people living alone within the community had difficulties finding medicines that had been dropped, while one healthcare professional further highlighted that these patients often do not report these difficulties. Management techniques included some participants taking a new tablet, which can result in medication running out earlier than prescribed.
“Half the time I probably think I’ve put two in my mouth, and I probably only put one and lost one on the floor”(P4)

#### 3.2.2. Dimensions and Scoring to Improve Handling

A significant number of the difficulties associated with handling tablets were related to small, round tablets, and healthcare professionals highlighted that, in their experience, these difficulties were present regardless of age. There was a general preference, therefore, for oval-shaped tablets and the caplet shape, which healthcare professionals highlighted would be easier to handle. Informal carers agreed and had a preference for tablets with a “pillow shape”:
“And I think if they can make them so that they don’t roll all over the place if you drop them. And those are the tinier ones obviously, they’re the flat-sided ones. The pillow ones, they’re fine, it’s just the tiny ones seem to roll” (C4)

Preferences for the oval and caplet shapes were often largely related to their relative thickness in comparison to the 6 mm and 7 mm round tablets. Participants described the need for tablets to be “chunkier” due to the loss of fine finger movement in old age. The 12 × 7 mm oval, for example, was preferred over the 10 mm round due to the thickness and ease of picking when on a flat surface. However, the majority of participants referred to the difficult compromise between ensuring tablets were easy to handle and the swallowability (Theme 3). Healthcare professionals further highlighted that within secondary care and within care homes, they tend to almost “spoon feed” patients; therefore, smaller tablets may be more appropriate, whereas in their own homes’ poor manual dexterity or poor eyesight is more of a concern. One healthcare professional suggested instead that tablets with an indentation are easier to handle and referred to the potential to use scoring as a “grip”:
“The scoring, yes, it’s a marker to actually cut the tablet, however it provides that grip that’s needed to control the tablet.” (HCP5)

### 3.3. Swallowability

#### 3.3.1. Dimensions

##### Size: A Balancing Act

The swallowability of the dosage form was a concern for older people living within care homes, secondary care and in the community. While, in general, smaller tablets aid swallowability, concerns were raised in relation to tablets that were too small (≤7 mm round), such as Amitriptyline. These preparations were difficult to swallow and informal and professional carers suggested this may be due to difficulties feeling this size tablet in the mouth. Older people agreed and reported difficulties determining whether it is “just the sensation” of the tablet being stuck in the throat or whether “it is actually there”.
“He talks about also, the small tablet, which I think is the rivaroxaban, he talks about that as being difficult for him to swallow and I wonder if it’s because he can’t really feel it in his mouth.” (C2)

Larger tablets may also be difficult to swallow. Preparations that were particularly difficult included calcium and vitamin D preparations, metformin, co-codamol, paracetamol and antibiotics (Appendix A). Management techniques deployed by patients included modifying tablets by breaking, repositioning the head when swallowing, drinking more water and sometimes “ignoring it” because “it’s going to go down eventually”. The need to deploy management techniques led to reduced acceptability of these preparations. The large size of some of these preparations further impacted the psychological willingness of patients to swallow the tablet, with a preference for smaller tablets which “don’t look as off-putting”. The potential for the large size to result in non-adherence led to some healthcare professionals prescribing multiple smaller tablets, adding to the overall pill burden for the patient.
“And then when you get to these, this bigger size, (18 × 7 mm caplet) it’s more about, you know, the thought of trying to swallow that bigger size can sometimes put patients off” (HCP8)

##### The Relationship between Size and Shape

Both older people and healthcare professionals described a relationship between size and shape when considering the difficulties swallowing some tablets. Difficulties swallowing large round (>10 mm) preparations, including co-codamol and paracetamol, could be overcome by changing to a “bullet” or caplet shape (Appendix A), and there was also a general preference for the oval shape over the round. However, the thickness and size of these shapes are important; informal carers voiced their concern over the thickness of both the placebo caplet and the 12 × 7 mm oval, and suggested that statin tablets that had the same length and width of the 12 × 7 mm oval, but that had a reduced depth, were easier to swallow as they were “not as thick”. Larger caplet and oval shapes (including the 16.5 × 8.9 mm oval) needed to be broken in half by both older people and healthcare professionals prior to administration. Social carers highlighted that the modification of tablets can present their own challenges, such as the potential to crumble due to a poorly functioning score line, leading to reduced acceptance and unintentional non-adherence to the prescribed dose. The time taken to modify doses was also a major concern for care professionals, whereas older people and carers placed a greater focus on the reduced acceptability of these preparations due to the difficulties swallowing.
“Do you know what I used to do, I used to break them into four and that was horrendous having to do” (P4)

#### 3.3.2. Palatability: Coating, Texture, Mouthfeel and Taste

The coating of tablets was described by the majority of participants as being a helpful tool to aid swallowability. Participants referred to the ease of swallowing tablets that appear “shiny”, have a “sugar coating” and that are “slippery”. However, there was also a need to consider the visual aesthetics of the coating, especially for patients with ill mental health; one patient in a nursing home (caring for those with mental illness) refused ibuprofen with a bright pink sugar coating due to a dislike of brighter colours. This led to the additional need to ensure the correct brand of ibuprofen was ordered in from the pharmacy. Care professionals therefore suggested white may be a more appropriate colour for these patients, highlighting the importance of considering coating alongside visual identification (Theme 1). One care professional highlighted the potential conflict that occurs when patients receive a formulation that is coated but that is unusually coloured:
“If I’ve not tasted it, I wouldn’t have known that the purpose of you making that coating is for me to have the sugary part of it … You say, oh try, it’s nice taste but you are trying to convince me to take it, but my eyes have already said that is not right, why is it pink instead of white?” (HCP24)

In contrast, uncoated tablets had a negative impact on texture of the dosage form and a subsequent effect on mouthfeel. Some tablets were described as having the potential to disintegrate in the mouth prior to swallowing, reducing acceptability of these preparations amongst informal carers due to concern that an incomplete dosage was being administered. Uncoated tablets were further described as being “chalky” and “powdery”, which again impacted acceptability and had a subsequent impact on taste and swallowability.
“And they’re not coated so they’re chalky you know; you need a good swallow of water to be able to get them down” (P7)

Participants had a preference for the plastic texture and mouthfeel of capsules over uncoated tablets; however, the large size of some capsule preparations (described as similar to the 18 × 7 mm caplet) led to difficulties swallowing, highlighting the need to consider both palatability and the dimensions to optimise swallowability. Furthermore, capsule formulations containing gelatine require careful consideration; healthcare professionals highlighted that patients often refuse these preparations due to religious beliefs, leading to intentional non-adherence.

In addition to optimisation of mouthfeel, the sugar coating was highlighted by participants as a tool to improve taste and, therefore, acceptance. Bitter tasting formulations, including paracetamol and phosphate preparations, were described as being more difficult to take. Healthcare professionals further highlighted difficulties associated with calcium and vitamin D preparations due to the taste, leading to intentional non-adherence, which is often not picked up until annual medication reviews. Some older people reported a preference for the formulation not to taste at all, whereas others described the potential to use a “neutral flavour” to help improve acceptance. The bitter taste was further described as a potential tool to aid memorability of taking a tablet. There is, therefore, a need for palatability to be considered alongside the dimensions, appearance and individual needs of each patient.
“If people were, didn’t like taking medication, if there was some neutral but pleasant flavour that they might have … it would have to be something very neutral, because obviously different people like different things” (P18)

## 4. Discussion

### 4.1. Main Findings

This study found that the formulation of oral solid dosage forms has an impact on an older person’s ability to identify, handle and swallow oral solid dosage forms. Figure 2 illustrates the relationship between key characteristics and each stage of the medication taking process. The characteristics can be classified into three main categories: dimensions, appearance and palatability [15]. The dimensions have an impact on all stages; in general, small round tablets (≤7 mm) are least accepted amongst older people and their carers and were perceived to have a negative impact on all stages. Interestingly, this includes swallowability due to the need to sense the tablet in the mouth; there is, therefore, a need to consider mouthfeel alongside the dimensions to optimise this stage of the medication taking process. Preparations with two colours are especially easy to identify and are more memorable, as are those with a more distinctive shape. Markings are generally used as a last resort to aid medication identification, and the presence of a score line can further aid handling; however, this requires further investigation. Palatability includes aspects such as the coating and taste, and while previous studies have identified the importance of this on improving swallowability [22], the current study further highlights the importance of these characteristics to aid medication identification and memorability. Several other factors also determine preferences for formulation characteristics, as summarised in Figure 3, and there is a need to ensure these are also considered when designing an older person’s patient centric drug product.

### 4.2. Comparison to Other Studies

Medication identification is complex for older people who are taking a greater number of medications [23]. Studies in polypharmacy patients (aged 29–80 years) have highlighted the importance of formulation characteristics for these patients; the use of white tablets was described as “boring” and innovative shapes such as a heart for cardiovascular medication can further improve medication recognition [24]. Older people are also more likely to suffer from visual impairment [25] and, therefore, commonly recognise oral medications based on their size, shape, colour and embossing, rather than reading the product label [4]. In particular, tablets with two colours result in faster detection times [26]. This study supports these findings and further highlights the importance of colour alongside coating and taste to make tablets easy to identify and memorable. However, changes in appearance that arise as a result of different brands and generics can significantly increase the odds of non-persistence, which may have important clinical implications [27]. Advice from GPs and pharmacists is key to improve the use of these tablets [28]; however, the present study highlights that changes are often not explained to patients, which can result in increased anxiety and reduced acceptance.

The dimensions also have a significant impact on the removal of medication from packaging and handling. Findings from the study highlight the difficulties associated with small round tablets (≤7 mm), especially due to the flat side, which can make picking difficult. Previous studies in the general adult population (aged 18–45 years) have used 3D printing to create a tilted diamond shape in order to tackle dexterity issues; the shape remains on a tilted position while on a flat surface, thereby allowing for easy picking [29]. However, despite being specifically designed for ease of picking, this shape scored lowest during the picking session [29], highlighting the need to involve patients when optimising dosage form characteristics. Healthcare professionals in the present study discussed the potential to assess the patient’s ability to handle medication prior to dispensing. However, the findings highlight the need for this to be broadened to include informal carers, who are often responsible for handling and administering medication. Informal carers often find the role of medication management challenging [30], and the present study highlights some difficult decisions they need to make as part of the medication administration process, such as asking for alternative brands due to the potential for non-adherence. In order to ensure they receive the most appropriate formulation and minimise caregiver burden, there is a need for collaborative partnerships between informal carers and healthcare professionals [9] and this must include exploring issues in relation to characteristics of the dosage form.

The use of larger tablets to aid handling requires a balance alongside the swallowability. Previous studies in healthy adult males (aged 24–33 years) have found that swallowing larger tablets (greater than 8 mm) requires significantly more swallows and more effort than smaller tablets [31]. Patients use management techniques to overcome difficulties, including drinking more water, splitting/crushing tablet or mixing it with food [32]. Similar management techniques were reported in this study; however, this can affect the stability, safety and efficacy of the drug, especially for modified release preparations [33]. Consideration of the shape may help to overcome some of these challenges, as the oesophageal transit of tablets is dependent on both shape and size [20,34]. Oval shapes have a faster oesophageal transit time compared to round tablets [20], and the study supports older peoples’ preferences for oval and caplet shapes in comparison to large round formulations of paracetamol and metformin. However, this does raise a further issue concerning cost, as oval paracetamol tablets may be more expensive than round paracetamol tablets, whereby community pharmacies may not be remunerated for dispensing the more expensive medication; this further underlines the need for a collaborative approach between patients, carers, healthcare professionals and healthcare providers [9].

Smaller tablets (≤7 mm) can also present their own challenges. These tablets are difficult to feel in the mouth and can therefore lead to the perception they have not been completely swallowed, highlighting the importance of mouthfeel when optimising swallowability. Similar findings were reported in a study investigating patients’ willingness to pay for oral solid dosage forms; older patients were less negative about larger-sized tablets, partly due to the difficulties seeing and swallowing smaller-sized pills [35]. Difficulties swallowing as a result of tablets being “too small to sense” have also been reported within the general adult population [36]. Optimisation of swallowability with a sole focus on the dimensions of the dosage form is therefore challenging, and the present study highlights the need for a more holistic approach, taking into account the palatability to overcome some of these difficulties.

Palatability has been defined as “the overall appreciation of a (often oral) medicine by organoleptic properties, such as vision (appearance), smell, taste, aftertaste and mouth feel and possibly also sound (auditory clues)” [5]. Optimisation of taste, mouthfeel and texture can all be achieved through the use of a coating; uncoated tablets require more water to swallow, take longer to swallow and cause tablets to lodge within the oesophagus [22]. The results in this study support these findings, with older people in general highlighting the superior mouthfeel, texture and swallowability of coated preparations. Preference for taste is often more personal and can differ between individuals; a previous study (in participants over 18 years of age, *n* = 300) found that while the majority (55%) had a preference for a dosage form that did not taste, some (40.7%) preferred a sweet taste while others (4.3%) had a preference towards a bitter tasting tablet [37]. Similar results were found in the present study, with preferences for taste varying between participants; patients taking a large number of tablets described the potential for taste to improve memorability, while those within care homes had a preference towards no taste at all. A greater priority should, therefore, be placed on the use of coated tablets for older people self-managing multiple medications to help improve memorability. This also has the potential to aid identification, as the use of a coloured coating can also further differentiate between tablets.

### 4.3. Strengths and Limitations

The qualitative design of this study was based on findings from a systematic review, which highlighted the paucity of qualitative research in this area [15]. The study involved older people, informal carers and health and social care professionals, providing the perspectives of all those involved in an individual’s therapy. The use of placebo tablets provided participants with a point of reference to communicate their ideas. Purposeful sampling was used to target participants from a range of ethnicities; however, these participants were particularly difficult to recruit, despite the lead researcher having a South Asian heritage. Older people unable to provide informed consent were excluded, although swallowing difficulties are commonly seen in patients with cognitive changes [38]. The study did, however, include the views of health and social care professionals who are often responsible for administering medication to these patients.

### 4.4. Future Work

The present study highlights the impact of external factors (Figure 3) on preferences for formulation characteristics and future work should aim to explore these in further detail. Further exploratory work on the impact of conditions such as dementia is particularly important, due to the progressive nature of these diseases, leading to a shift in responsibility of medication management to informal carers [30]. Further work is also needed in Black and Minority Ethnic groups; future work can help determine whether factors such as language barriers may result in greater emphasis being placed on certain areas of the formulation, such as the appearance. In addition, while the current study focused on traditional oral solid dosage forms, further work is needed to explore the use of alternative dosage forms, such as orodispersible tablets, to improve patient acceptability, while previous studies both within paediatrics and the general adult population have explored the use of 3D printing to improve acceptance [29,39]. The authors believe that 3D printing has the potential to be a particularly valuable tool when designing patient centric drug products for older people, as an individualised approach is necessary to take into account the heterogeneous nature of this population. Further work should explore the use of this technology in the older population, taking into account the recommendations made in this study. In order to ensure patient centric medicine design, future work in this area should also include working in collaboration with the pharmaceutical industry, so that recommendations can be incorporated into the Quality Target Product Profile for specific drug products.

## 5. Conclusions

Medication adherence in older people is challenging, and a key determinant of this is acceptability. Characteristics of oral solid dosage forms are key for ensuring acceptability and, therefore, adherence. There is a need to consider the medication taking process as a whole when optimising these characteristics. Tablets must be visually appealing and memorable, be easy to handle and have optimum swallowability (by considering the dimensions and palatability side by side).

While there is no “one size fits all” approach, the results highlight a trend towards some key characteristics to optimise these stages of the medication taking process. Smaller tablets (≤7 mm) should generally be avoided and, where necessary, other characteristics such as colour and mouthfeel should be optimised to aid identification and swallowability. The results further highlight a preference towards brightly coloured and two-colour preparations and the potential to use the appearance of medication to categorise tablets based on their therapeutic indications or the time needed to be taken. This is especially important for older people self-managing multiple medications. Tablets should be easy to handle and the use of a score line or other innovative shapes that tackle dexterity issues in this population requires further investigation. The use of a coating is preferred among older people and can help to optimise both medication identification and swallowability. In order to ensure the most appropriate formulation is dispensed, health and social care professionals have a key role in assessing the suitability of a formulation and collaborating with both older people and informal carers responsible for the medication administration process.

Environmental, patient, medication and disease characteristics may lead to a greater emphasis being placed on certain stages of the medication taking process, and these factors therefore also determine preferences for formulation characteristics. Overall, developing an age appropriate dosage design for older people requires a holistic, patient centric approach to improve acceptance and adherence.

## Figures and Tables

**Figure 1 pharmaceutics-12-00905-f001:**
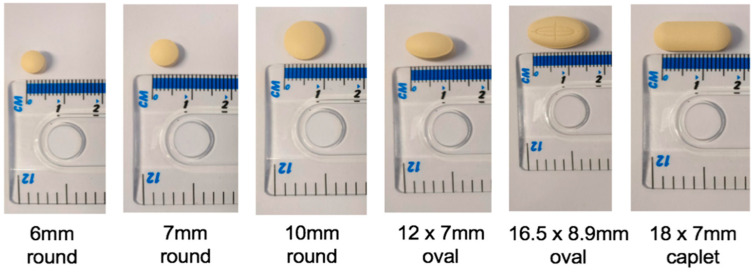
Placebo tablets used in interviews.

**Figure 2 pharmaceutics-12-00905-f002:**
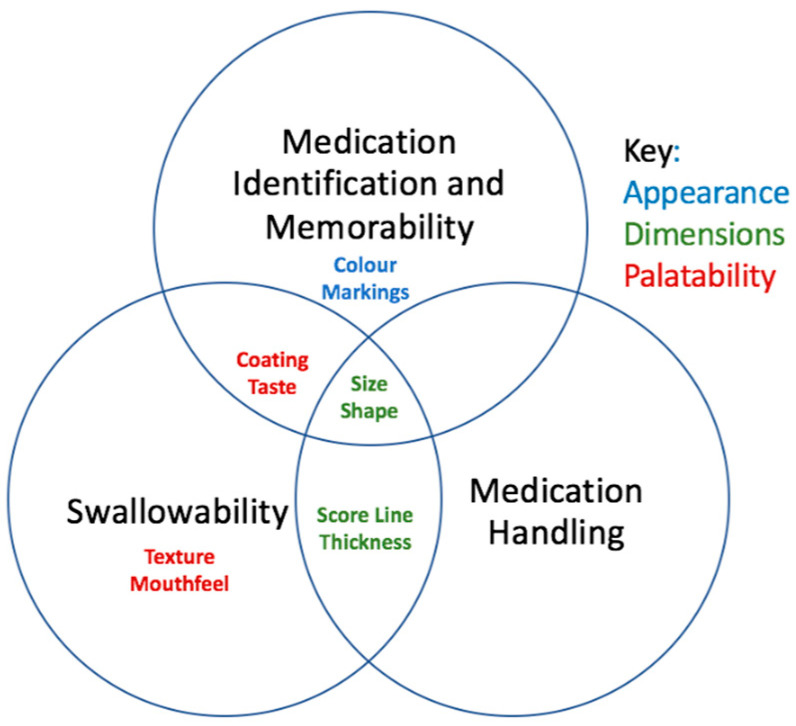
The relationship between key characteristics and each stage of the medication taking process.

**Figure 3 pharmaceutics-12-00905-f003:**
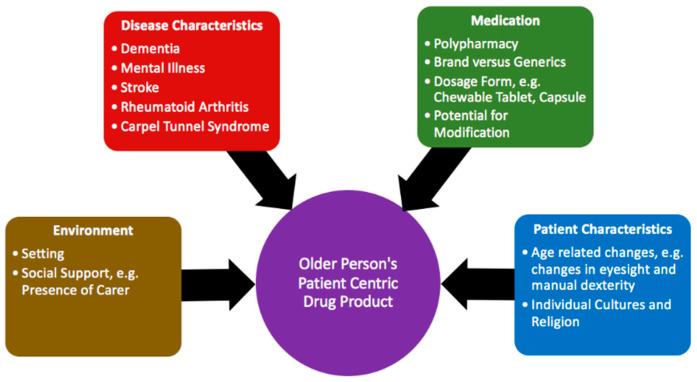
A holistic approach towards the design of an older person’s patient-centric drug product—four key areas have an impact on preferences for formulation characteristics.

**Table 1 pharmaceutics-12-00905-t001:** Summary of the topic areas explored by the semistructured interview schedule.

**Older People**
Background Information
Details regarding Current Medication
Impact of Characteristics on ability to take Medication as directed
Changes to help make Medication easier to take
**Carers**
Background Information
Details regarding Current Medication administered
Impact of Characteristics on ability to administer Medication as directed
Changes to help make Medication easier to administer
**Health and Social Care Professionals**
Background Information
Experience with Older People/their Carers, their Medication and the Impact of the Physical Characteristics
Changes to help make Medication easier for older people or their carers to take/administer
Role of Health and Social Care Professionals in relation to Optimising the Physical Characteristics of Medication

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
