# Peer review of "Patient-Centric Medicine Design: Key Characteristics of Oral Solid Dosage Forms that Improve Adherence and Acceptance in Older People"

_pharmaceutics, 2020, doi:10.3390/pharmaceutics12100905_

Round 1
Reviewer 1 Report
This study provides some suggestions regarding the design of tablets
suitable for elderly patients, for the improvement of therapeutic
adherence.
The advice comes from a survey carried out with some elderly patients and
care givers. Although the work is qualitative in nature, it provides interesting
points for reflection.
In my opinion, I think that it is appropriate for the publication.
Some of my suggestions or comments are reported below:
1) The survey was conducted on a very low number of respondents.
2) Although a qualitative study was reported, some statistical
informations would have given more prominence to the analysis.
3) Reporting the results achieved with graphs (pie, radar histograms)
, etc.) could improve the presentation of the data.
4) Finally, you could have asked to interviewed which formulation to replace the tablets they would have gladly chosen
I think with minimal revisions the work can be published.
Best Regards
Author Response
Response to Reviewer 1 Comments
The authors would like to thank the reviewer for their comments. Please see responses to comments in red below- specific changes that have been made to the document have been highlighted in bold to aid review. (These have also been made as tracked changes on the article itself).
1) The survey was conducted on a very low number of respondents.
Response 1:
Qualitative studies in general tend to have smaller sample sizes compared to quantitative studies to enable the collection of rich, in-depth information that is essential for this type of study. Previous studies have found that qualitative study sample sizes of as small as 12 can be enough to reach data saturation- the point at which no new data themes emerge (1). This is, however, dependent on the sample characteristics. Due to the importance of ensuring a representative sample, participants in a range of settings were interviewed including the community, care homes and secondary care. Data collection continued until data saturation was reached, resulting in a total of 52 interviews- an upper limit for qualitative research. By adopting this approach, the authors were able to ensure the appropriateness of the sample composition and collect detailed information to provide a valid response to the research question. This is especially important for qualitative research, where the composition of the sample is crucial and purposive sampling is usually used in order to select information rich cases (2).
References:
1) Guest, G., Bunce, A., & Johnson, L. (2006). How Many Interviews Are Enough?: An Experiment with Data Saturation and Variability. Field Methods, 18(1), 59–82. https://doi.org/10.1177/1525822X05279903
2) Vasileiou, K., Barnett, J., Thorpe, S. et al. Characterising and justifying sample size sufficiency in interview-based studies: systematic analysis of qualitative health research over a 15-year period. BMC Med Res Methodol 18, 148 (2018). https://doi.org/10.1186/s12874-018-0594-7
2) Although a qualitative study was reported, some statistical informations would have given more prominence to the analysis.
Response 2:
The following statistical information has been added:
The mean age of older people: 78.7
The age range of older people: 66-97
The mean age of carers: 60.9
The age range of carers: 48-70
(pg. 4 lines 152-153).
As this was a qualitative study, further statistical analysis is difficult as the authors analysis centred around providing a more detailed understanding of the thoughts and feelings of research participants. The previous systematic review on this topic found that qualitative studies in this area are scarce and therefore one of the main aims of this study was to take a purely qualitative approach to help us understand preferences for oral solid dosage forms, specifically why certain characteristics were preferred and the context in which answers were provided. Understanding the thoughts of participants on this topic area will allow the authors to conduct future quantitative research in which the recommendations are implemented and a more statistical approach to the analysis undertaken.
3) Reporting the results achieved with graphs (pie, radar histograms), etc.) could improve the presentation of the data.
Response 3:
Unfortunately, due to the qualitative nature of this study, presentation of the results using graphs would be difficult. Qualitative studies in general aim to provide an insight into participants’ thoughts and feelings. The analysis method is a complex process of transcribing and checking interview transcripts, reading between the lines, coding the data and generating themes to present the data in a coherent and meaningful way. The questions used in the interview guide were semi-structured and open ended to limit bias and collect rich data, however this also makes quantitative analysis of the data difficult. Instead, the authors provided quotations in the results section with an aim to provide a voice to the experiences of participants and provide evidence for each theme that was developed. The authors also aimed to summarise the key findings in a Venn diagram to provide a visual representation of the impact of formulation on the medication taking process (Figure 2, page 10). This provides a representation of the complexities associated with designing an older person’s patient centric drug product and the importance of considering the medication taking process as a whole.
4) Finally, you could have asked to interviewed which formulation to replace the tablets they would have gladly chosen
Response 4:
This is an important topic area, and the authors aimed to address this question in the fourth section of the interview guide, in which older people and carers were asked about what changes would help make their medication easier to take/administer (Table 1, pg. 3). Health and social care professionals were also asked about changes that would help to make medication easier for older people to take or their carers to administer (Table 1, pg. 3). In addition to this dedicated section of the interview schedule, the lead researcher made sure to ask follow up questions based on the responses given by participants. For example, when discussing section 2 of the interview schedule (which explored the impact of formulation characteristics on the ability to take/administer medication), participants were given the opportunity to refer to specific medicines that were easier or harder to take. This gave participants the opportunity to discuss which formulations they prefer in their own words and this can be illustrated using quotes in the results section; for example, pg. 5, line 208-211: “Now, these particular ones, without opening it, I think are only white, though I have had some that are white and red, and that, actually, would be a lot better because then I would remember better, I think, when it comes to relating what I've done with the Paracetamol… if there was a query in my head I might be able to remember better if it had more of a distinctive combination." (P3). This quote illustrates the participant having a preference for formulations with two colours rather than having just white tablets and therefore provides evidence of preferences for alternative formulation.

Reviewer 2 Report
The article aimed to investigate the characteristics of oral solid dosage forms that affect their acceptance in older people. A qualitative approach was applied. The topic was interesting, and the article is a sort of proof-of-concept, based on real-world data, of the approach reported in the latest reflection papers of regulatory authorities such as EMA. However, some majors should be addressed by the authors. Firstly, the title seems not coherent with the aims of the study. Indeed, the current title suggests a more formulative approach for improving the adherence and acceptance, whereas the manuscript is more focused on collecting data about the most critical attributes of tablets and other oral dosage forms from patients, caregivers and healthcare professionals involved in the assistance of the elderly. Secondly, the study design presents some criticisms. It is not clear how the sample size was defined. No information was available about the clinical history of the enrolled older people. Only age, gender and ethnicity were reported. For example, difficulties in swallowability can be due to the dosage forms, but they can be also due to a reduced patient capability to swallow (dysphagic patients). Considering the limited number of older people (n=18), some bias can be present. Finally, the authors provide suggestions based on the results of qualitative analyses. However, real-world data and the overall results did not provide demonstrations of the impact on patient’s adherence. Maybe, the aims should be reconsidered.
Author Response
Response to Reviewer 2 Comments
The authors would like to thank the reviewer for their comments. Please see responses to comments in red below- specific changes that have been made to the document have been highlighted in bold to aid review. (These have also been made as tracked changes on the article itself).
1) Firstly, the title seems not coherent with the aims of the study. Indeed, the current title suggests a more formulative approach for improving the adherence and acceptance, whereas the manuscript is more focused on collecting data about the most critical attributes of tablets and other oral dosage forms from patients, caregivers and healthcare professionals involved in the assistance of the elderly.
Response 1:
The authors acknowledge the suggestion of a more formulative approach with the current title, and have therefore changed the title in line with the reviewer’s suggestion to: “Patient Centric Medicine Design: Key Characteristics of Oral Solid Dosage Forms that Improve Adherence and Acceptance in Older People.”
2) Secondly, the study design presents some criticisms. It is not clear how the sample size was defined.
Response 2:
The sample size when considering qualitative studies is most commonly determined by data saturation. This has been defined as the point at which “no new information or themes are observed” (page 2, lines 86-87). Previous research in this area has referred to data saturation as the “gold standard by which purposive samples are determined” (1) and therefore a failure to reach saturation can impact the quality of the research.
In order to clarify that this was the way in which the sample size was defined, the authors have updated the above passage to: “The sample size was determined by data saturation and this has been defined as the point at which “no new information or themes are observed.” This was therefore the key criterion for discontinuing data collection.” (Pg. 2 lines 105-107)
References:
1) Guest, G., Bunce, A., & Johnson, L. (2006). How Many Interviews Are Enough?: An Experiment with Data Saturation and Variability. Field Methods, 18(1), 59–82. https://doi.org/10.1177/1525822X05279903
3) No information was available about the clinical history of the enrolled older people. Only age, gender and ethnicity were reported. For example, difficulties in swallowability can be due to the dosage forms, but they can be also due to a reduced patient capability to swallow (dysphagic patients).
Response 3:
Older people were not specifically separated by whether they suffered from dysphagia as the overall aim of this study was to explore general preferences for characteristics of oral solid dosage forms amongst older people and their carers. However, where specific conditions had an influence on the results, the authors aimed to document and provide evidence of this. For example, brighter colours and the use of colour to improve memorability was particularly important for people with dementia and this was referred to on pg. 4 lines 172-173. Specific conditions were also responsible for difficulties handling tablets and these have been referred to on pg. 6, lines 281-283.
4) Considering the limited number of older people (n=18), some bias can be present.
Response 4:
Qualitative studies in general tend to have smaller sample sizes compared to quantitative studies to enable the collection of rich, in-depth information that is essential for this type of study. Previous studies have found that qualitative study sample sizes of as small as 12 can be enough to reach data saturation- the point at which no new data themes emerge (1). This is, however, dependent on the sample characteristics. Due to the importance of ensuring a representative sample, older people in a range of settings were interviewed including the community, care homes and secondary care. By adopting this approach, the authors were able to ensure the appropriateness of the sample composition and collect detailed information to provide a valid response to the research question. This is especially important for qualitative research, where the composition of the sample is crucial and purposive sampling is usually used in order to select information rich cases (2).
References:
1) Guest, G., Bunce, A., & Johnson, L. (2006). How Many Interviews Are Enough?: An Experiment with Data Saturation and Variability. Field Methods, 18(1), 59–82. https://doi.org/10.1177/1525822X05279903
2) Vasileiou, K., Barnett, J., Thorpe, S. et al. Characterising and justifying sample size sufficiency in interview-based studies: systematic analysis of qualitative health research over a 15-year period. BMC Med Res Methodol 18, 148 (2018). https://doi.org/10.1186/s12874-018-0594-7
5) Finally, the authors provide suggestions based on the results of qualitative analyses. However, real-world data and the overall results did not provide demonstrations of the impact on patient’s adherence. Maybe, the aims should be reconsidered.
Response 5:
The aim of the study was to “investigate the characteristics of oral solid dosage forms that will contribute to an age appropriate dosage design, with the aim to improve overall medication adherence and acceptance in older people” (pg. 2, lines 84-86). The focus on adherence and acceptance was determined by reference to the EMA reflection paper on the pharmaceutical development of medicines for older people, which states: “Patient acceptability can be defined as the ability and willingness of a patient to self-administer, and also of any of their lay or professional caregivers, to administer a medicinal product as intended. Patient acceptability is likely to have a significant impact on patient adherence, which can e.g. have an impact on the patient and caregiver quality of life.” It is therefore important to consider both of these outcomes when investigating the pharmaceutical development of medicines for older people. In order to clarify this, the authors have updated the paper to provide a summary of the above, including both a definition of adherence and acceptance as appropriate for this study:
“When discussing these characteristics, both the EMA and FDA refer to “patient acceptability” [3,4], defined as “an overall ability of the patient and caregiver to use a medicinal product as intended” [5]. Acceptability can have a significant impact on adherence, defined by NICE as “the extent to which the patient's action matches the agreed recommendations” [6].” (Pg. 1, Line 40-43)
In terms of the results of the qualitative analysis, the authors have aimed to link the results back to patient adherence and acceptance throughout the analysis. These links have been made by considering the definitions of acceptance and adherence above that have been added on pg. 1 lines 40-43. Please find below some examples of references to adherence and acceptance that have been made in the qualitative analysis in order to ensure that the results relate back to the aims of the study:
- a) pg. 4, lines 187-189: “overall understanding and, therefore, adherence could be improved if colour could be used for groups of medications when counselling patients.”
- b) pg. 5, lines 221-223: “Informal carers took on the responsibility of weighing up the importance of the tablet and discussed the need to ask for an alternative from the doctor if “more important” medications were of a small size to ensure patient adherence.”
- c) pg. 5, lines 243-245: “Informal carers, in particular, highlighted this would improve acceptance, as it would reduce the worry when administering two tablets which look similar.”
- d) pg. 6, lines 272-273: “Informal carers described the need to double check that all medications were taken, due to the likelihood of unintentional non-adherence”
- e) pg. 8, lines 353-354: “The potential for the large size to result in non-adherence led to some healthcare professionals prescribing multiple smaller tablets”
- f) pg. 8, lines 371-373: “the modification of tablets can present their own challenges, such as the potential to crumble due to a poorly functioning score line, leading to reduced acceptance and unintentional non-adherence to the prescribed dose.”

Reviewer 3 Report
A brief summary:
The aim of the study was to investigate the characteristics of oral solid dosage forms that contribute to an appropriate design for older patients. 52 interviews were conducted with older people, informal (family) carers and health and social care professionals. Findings: size, shape, colour, coating and taste had an impact on identification and memorability, medication handling and swallowability.
Broad comments:
This reviewer thinks that the issue of patient friendly drug delivery is important and worth raising awareness about. Using a qualitative approach involving end users seems appropriate. However, the approach taken in this study was very broad: 52 interviews is a lot and the inclusion criteria are broad: patients over the age of 65, without any further differentiation, for example with regards to age, physical state, amount of medications etc. This is actually not in line with the paper´s own introduction, stating: The design of a patient centric drug product for the older population, however, presents a challenge due to the heterogeneous nature of this population [4]. Co-morbidities and age-related differences, including changes in cognition, motor functions and sensory functions, need to be considered. (line 42-44). It seems like this broad approach also influenced the results, which were, unfortunately, self-evident at time. One example is the following: In contrast, older people living alone within the community described the importance of using bright colours to ensure tablets are easily visible, especially when they were accidentally dropped on the floor. (line 141 – 142). It would probably be more interesting to investigate and provide more detail related to this (for example correlate brightness to eye vision, or even illumination) and similar issues (for example the effect of dysphagia and/or how many tablets can be managed before losing track). Details like this, which is probably more useful when the findings are to be used in practical life, were lost by the broad design of the study.
This lack of specificity can also be seen in the abstract, for example line 22-25: Optimising the dimensions can further make tablets easy to handle, while both the dimensions and palatability have a key role in ensuring optimum swallowability. Environmental, patient, medication and disease characteristics also determine preferences for formulation. Developing an age appropriate dosage design for older people, therefore, requires a holistic, patient centric approach to improve adherence and acceptance. It is hard to disagree with any of this, but it is also hard to extract any new, meaningful information about what should be done and what should change, to benefit the patients. Keeping in mind that the authors have interviewed 52 persons, and done extensive work in this field before (as reflected in the reference list), a suggestion of more concrete recommendations was maybe anticipated.
Specific comments:
The paper is focusing only on “normal” disintegrating tablets, which is not reflected in the heading.
Introduction: Interesting references are used, but it would be nice if more details and examples could be provided, for example here (line 54-56): The lack of appropriate licensed dosage forms for this population results in dosage forms being routinely modified [8]. These alterations can further complicate treatment by potentially changing the bioavailability, toxicity and stability of medicines [9].
Method: line 74/75: this reviewer doesn´t know what this means. The “interview guide” consists of 4 topics. This reviewer is therefore uncertain whether this in fact can be classified as a “semi-structured interview”.
Results: The results are clearly presented and separated according to source (patient, carer, professional). This reviewer thinks that particularly some of the input provided by health care personnel was new and interesting.
Line 329 – 332: However, the thickness and size of these shapes are important; informal carers voiced their concern over the thickness of both the placebo caplet and the 12 x 7 mm oval, and suggested that statin tablets that were similar in size to the 12 x 7 mm oval but “not as thick” were easier to swallow. What does this mean: “similar in size, but not as thick”: that they were longer, or what?
Line 354-356: does this quote stem from a patient or health care worker?
Discussion:
This reviewer thinks that the discussion is too superficial. For example (line 506-510): Similar results were found in the present study, with preferences for taste varying between participants; patients taking a large number of tablets described the potential for taste to improve memorability, while those within care homes had a preference towards no taste at all, highlighting the need for a personalised approach when considering this aspect of palatability. It is unclear to the reader how the authors pictures that this issue is to be resolved?
This reviewer thinks it would be nice if the discussion could address issues like the following:
From the results, what was new and maybe even surprising?
Which new insight was provided by the carers and health care practitioners? (a request inspired by the paper´s introduction: However, a recent systematic review published in this area found only a single study that involved the views of General Practitioners and no studies that involved formal (paid) or informal (family) carers [12].)
How can industry use the results already now, and which issues need to be further scrutinized?
For example, what is the best way to use colour? (The results provided many suggestions, so a prioritization is maybe required.)
Suggestions on how to handle the issue of different brands and generics?
It would also be nice if the authors could help the readers with identifying critical medical problems from those that seem manageable or even mostly related to personal taste.
For example:
Line 470 – 472: Informal carers often find the role of medication management challenging [26] and the present study highlights some difficult decisions they need to make as part of the medication administration process, such as asking for alternative brands due to the potential for non-adherence. For the reader, the existence of alternative brand may seem like a reasonable solution.
Line 348-350: However, there was also a need to consider the visual aesthetics of the coating, especially for patients with ill mental-health; one patient in a nursing home (caring for those with mental illness) refused ibuprofen with a bright pink sugar coating due to a dislike of brighter colours. Is this regarded as a medical problem, keeping in mind that ibuprofen is available as different types of tablets (brands) and also other formulations?
Line 310-314: Larger tablets may also be difficult to swallow. Preparations that were particularly difficult included calcium and vitamin D preparations, metformin, co-codamol, paracetamol and antibiotics (Appendix A. Table A2). Paracetamol is available as different types of tablets (brands) and also other formulations. Calcium and vitamin D preparations are mainly chewing tablets.
Line 313 – 314: Management techniques deployed by patients included modifying tablets by breaking, re-positioning the head when swallowing, drinking more water and sometimes “ignoring it” because “it’s going to go down eventually.” & line 365 -366: “And they’re not coated so they’re chalky you know, you need a good swallow of water to be able to get them down” (P7). To the reader, this may seem like manageable strategies mainly resolving the issues.
Line 371 – 373: Furthermore, capsule formulations containing gelatine require careful consideration; healthcare professionals highlighted that patients often refuse these preparations due to Religious beliefs, leading to intentional non-adherence. Is this a medical problem? (Particularly keeping in mind that most capsules are both halal and kosher.)
Line 477 – 480: Patients use management techniques to overcome difficulties, including drinking more water, splitting/crushing tablet or mixing it with food [28]. Similar management techniques were reported in this study, with participants using characteristics such as the break mark to modify dosage forms; however, this can affect the stability, safety and efficacy of the drug [29]. For this reviewer, it is surprising that the practice of using break marks to break tablets is problematic. Maybe the authors can point to a specific preparation where the stability, safety and efficacy was compromised, encountered during their study?
This is neither to criticize the result nor the method per se, but a call to discuss the results more thoroughly. For example: keeping in mind the wide range of brands and alternatives for most APIs, and the various coping strategies suggested by patients and carers, are there some particularly critical unresolved medical issue/API for which the formulation should be urgently improved, that the authors would like to raise awareness about?
This reviewer thinks that alternative products/brands and formulations (like oral solutions, orodispersible tablets, transdermal patches, suppositories) have the potential to improve the drug management process for the elderly. The authors, however, point to another solution (line 532-534): Previous studies both within paediatrics and the general adult population have explored the use of 3D printing to improve acceptance [25,37]; further work should explore the use of this technology in the older population, taking into account the recommendations made in this study. It would be nice if the authors could elaborate on why they think 3D printing is the best solution.
Figure 2 is nice (however, as pointed out in the result section, colour can also affect swallowability), but Figure 3 provides information of such a wide and general character, that it can probably be omitted.
Conclusions: The conclusion is very broad without offering any specific findings nor suggestions, only generalizations. This reviewer has read the paper carefully several times, and seem to have identifies the following:
- Small round tablets (≤ 7 mm) are least accepted amongst older people
- Coating is preferred
- Two-colour preparations are preferred
- Healthcare professionals suggests to assess the patient’s ability to handle medication prior to dispensing
- Healthcare professionals uses score lines for grip
Appendix A, Table A2: The presentation of specific preparations is a positive (however, the reviewer´s thoughts on paracetamol and calcium/vitamin D preparations have already been presented). However, the authors should consider whether Table A2 could be shortened a bit.
Author Response
Response to Reviewer 3 Comments
The authors would like to thank the reviewer for their comments. Please see responses to comments in red below- specific changes that have been made to the document have been highlighted in bold to aid review. (These have also been made as tracked changes on the article itself).
Broad Comments:
1) This reviewer thinks that the issue of patient friendly drug delivery is important and worth raising awareness about. Using a qualitative approach involving end users seems appropriate. However, the approach taken in this study was very broad: 52 interviews is a lot and the inclusion criteria are broad: patients over the age of 65, without any further differentiation, for example with regards to age, physical state, amount of medications etc. This is actually not in line with the paper´s own introduction, stating: The design of a patient centric drug product for the older population, however, presents a challenge due to the heterogeneous nature of this population [4]. Co-morbidities and age-related differences, including changes in cognition, motor functions and sensory functions, need to be considered. (line 42-44).
Response 1:
The authors agree that the approach taken in the study was broad; however, this was an important aspect of the design of the study when considering previous research in this area. While regulatory papers such as that published by the European Medicines Agency do highlight the importance of considering age-related differences, there have been very few studies which provide real-world data on what factors need to be considered when formulating an older person’s patient centric drug product. The authors previously conducted a systematic review on this topic which highlighted that there have been no previous qualitative studies that directly investigate preferences for characteristics amongst older people, their carers and health and social care professionals. The design of the study was therefore kept broad in order to provide data on what factors influence preferences for oral solid dosage forms, with the results providing an important foundation on which future, more targeted, research can be conducted. For example, from the results, the appearance of the dosage form is particularly important for people with dementia, who require visible and appealing colours (pg. 4 lines 163-164) and therefore future research should focus more on how appearance can be used to improve adherence in this target group.
2) It seems like this broad approach also influenced the results, which were, unfortunately, self-evident at time. One example is the following: In contrast, older people living alone within the community described the importance of using bright colours to ensure tablets are easily visible, especially when they were accidentally dropped on the floor. (line 141 – 142). It would probably be more interesting to investigate and provide more detail related to this (for example correlate brightness to eye vision, or even illumination) and similar issues (for example the effect of dysphagia and/or how many tablets can be managed before losing track). Details like this, which is probably more useful when the findings are to be used in practical life, were lost by the broad design of the study.
Response 2:
The study aimed to provide evidence of preferences for characteristics of oral solid dosage forms from a range of participants in a range of settings. Lines 151-153, for example, provide a comparison of the differences in priorities when considering appearance amongst older people within the community, and those within care homes. The authors felt that it was important to stay close to the quotes provided by participants and therefore all of the analysis was centred around the interview transcripts. Participants were asked on more general terms about the impact of medication characteristics on their ability to take medication as directed and where specific healthcare conditions and contexts were found to have an impact on this, the authors aimed to provide evidence of this. For example, pg 6, lines 272-274: “Participants referred in particular to rheumatoid arthritis, stroke, neuropathy and carpal tunnel syndrome, all of which affected older peoples’ ability to remove and handle smaller tablets.”
The authors note however that details on how preferences for formulation characteristics may be impacted by age related changes are important points to consider, and have aimed to address these details in the discussion to provide a more contextual understanding of the results; for example, pg.10, lines 494-496 discuss the impact of visual deterioration in old age and the resulting identification of products based on their appearance.
The results from healthcare professionals were also centred around the interview transcripts, and interestingly their explanation of the importance of factors such as colour did differ in the level of detail they provided. For example, healthcare professionals did refer to the significance of visual deterioration and the authors aimed to provide evidence of this on pg. 4, lines 164-166: “Healthcare professionals also referred to the importance of brighter colours due to a decline in visual acuity as a result of macular degeneration or cataracts.” In order to provide more detailed evidence of this, the authors have updated the transcript to include the following quote: “Is there something that maybe needs to be brighter because they've got, maybe, macular degeneration or they've got problems with cataracts” (HCP11) (Pg. 4 Lines 168-169)
3) This lack of specificity can also be seen in the abstract, for example line 22-25: Optimising the dimensions can further make tablets easy to handle, while both the dimensions and palatability have a key role in ensuring optimum swallowability. Environmental, patient, medication and disease characteristics also determine preferences for formulation. Developing an age appropriate dosage design for older people, therefore, requires a holistic, patient centric approach to improve adherence and acceptance. It is hard to disagree with any of this, but it is also hard to extract any new, meaningful information about what should be done and what should change, to benefit the patients. Keeping in mind that the authors have interviewed 52 persons, and done extensive work in this field before (as reflected in the reference list), a suggestion of more concrete recommendations was maybe anticipated.
Response 3:
The authors note the need for more concrete recommendations in the conclusion and have updated the conclusion according to the reviewer’s comments- please see response 20 below. The conclusion in the abstract has also been updated:
Abstract: Older people represent a very heterogeneous patient population and are the major user group of medication. Age-related changes mean that this population can encounter barriers towards taking medicines orally. The aim of this study was to investigate the characteristics of oral solid dosage forms that contribute to an age appropriate dosage design with an aim to improve overall medication adherence and acceptance in older people. 52 semi-structured interviews were conducted with older people, informal (family) carers and health and social care professionals. Formulation characteristics impacted three stages of the medication taking process: 1) medication identification and memorability; 2) medication handling; and 3) swallowability. Small round tablets (≤ 7 mm) are least accepted amongst older people and their carers and had a negative impact on all stages. The use of bright, two-coloured preparations and interesting shapes improves identification and further aids memorability of indications and the timing of tablets. Palatability, while useful to enhance swallowability, also has an impact on the visual appeal and memorability of medication. Environmental, patient, medication and disease characteristics also determine preferences for formulation. Developing an age appropriate dosage design for older people, therefore, requires a holistic, patient centric approach to improve adherence and acceptance. (Pg. 1, Lines 18-23)
Specific Comments:
1) The paper is focusing only on “normal” disintegrating tablets, which is not reflected in the heading.
Response 1:
The title has been updated to reflect a focus on “oral solid dosage forms,” to: “Patient Centric Medicine Design: Key Characteristics of Oral Solid Dosage Forms that Improve Adherence and Acceptance in Older People”
2) Introduction: Interesting references are used, but it would be nice if more details and examples could be provided, for example here (line 54-56): The lack of appropriate licensed dosage forms for this population results in dosage forms being routinely modified [8]. These alterations can further complicate treatment by potentially changing the bioavailability, toxicity and stability of medicines [9].
Response 2:
The above has been updated to include further details and examples, and now reads:
“The lack of appropriate licensed dosage forms for this population results in dosage forms being routinely modified, with medications affecting the Central Nervous System being the most frequently modified [10]. While modifications are sometimes necessary to facilitate fractional dosing and to overcome swallowing difficulties, studies and guidelines recommend that modifications are best avoided due to the legal and clinical risks that may arise [11]. In particular, alterations can further complicate treatment by potentially changing the bioavailability, toxicity and stability of medicines [12].” (Pg. 2 Lines 71-76)
3) Method: line 74/75: this reviewer doesn´t know what this means. The “interview guide” consists of 4 topics. This reviewer is therefore uncertain whether this in fact can be classified as a “semi-structured interview”.
Response 3:
Table 1 provides a summary of the four key topic areas explored by the semi structured interview schedule. For example, the third topic area (defined as the impact of characteristics on ability to take medication as directed) was explored by asking the question: “How do things like the shape, size, colour and coating (we call these the physical characteristics) affect your ability to take the medication as directed by the doctor?” The authors note that the use of the title “interview guide” for the table is not appropriate as this was not the actual interview guide that was used, but a summary of the topic areas explored by the guide. The title has therefore been updated to: “Table 1: Summary of the topic areas explored by the semi structured interview schedule” (pg. 3 Lines 112)
4) Results: The results are clearly presented and separated according to source (patient, carer, professional). This reviewer thinks that particularly some of the input provided by health care personnel was new and interesting.
Line 329 – 332: However, the thickness and size of these shapes are important; informal carers voiced their concern over the thickness of both the placebo caplet and the 12 x 7 mm oval, and suggested that statin tablets that were similar in size to the 12 x 7 mm oval but “not as thick” were easier to swallow. What does this mean: “similar in size, but not as thick”: that they were longer, or what?
Response 4:
The informal carer in this case was referring to statin tablets that had the same length and width as the 12 x 7mm oval placebo tablet, but that had a reduced depth which made them thinner and therefore easier to swallow. The text has been updated to clarify this to: “However, the thickness and size of these shapes are important; informal carers voiced their concern over the thickness of both the placebo caplet and the 12 x 7 mm oval, and suggested that statin tablets that had the same length and width of the 12 x 7mm oval but that had a reduced depth were easier to swallow as they were “not as thick.” (Pg.8 Lines 368-369)
5) Line 354-356: does this quote stem from a patient or health care worker?
Response 5:
This quote stems from a health care worker who was trying to illustrate what a patient would think if they were given an unusual coloured tablet, and is therefore speaking from the patient’s point of view. To clarify this further, the following text has been added: “One care professional highlighted the potential conflict that occurs when patients receive a formulation that is coated but that is unusually coloured:” (Pg.8 Lines 391-392)
6) Discussion: This reviewer thinks that the discussion is too superficial. For example (line 506-510): Similar results were found in the present study, with preferences for taste varying between participants; patients taking a large number of tablets described the potential for taste to improve memorability, while those within care homes had a preference towards no taste at all, highlighting the need for a personalised approach when considering this aspect of palatability. It is unclear to the reader how the authors pictures that this issue is to be resolved?
Response 6:
The authors have updated the discussion section in line with the reviewer’s recommendations; please see responses below. In terms of lines 506-510, the aim of this statement was to provide evidence of the differences in priorities when considering patients taking a large number of medications, and those who had medications managed for them e.g. within care homes. As noted on pg. 11 lines 556-560, optimisation of taste can be achieved through the use of a coating and the results therefore highlight that the use of coated tablets should be a greater priority for those managing multiple medications. This has been added as follows: “Similar results were found in the present study, with preferences for taste varying between participants; patients taking a large number of tablets described the potential for taste to improve memorability, while those within care homes had a preference towards no taste at all. A greater priority should therefore be placed on the use of coated tablets for older people self-managing multiple medications to help improve memorability. This also has the potential to aid identification as the use of a coloured coating can further help differentiate between tablets.” (Pg. 12 Lines 568-571)
7) This reviewer thinks it would be nice if the discussion could address issues like the following:
From the results, what was new and maybe even surprising?
Response 7:
The authors have aimed to summarise the main findings in the first part of the discussion (lines 392-404). However, the authors note the need to highlight the interesting new findings, which we have identified as follows:
- Preparations with two colours are especially easy to identify and are more memorable
- Markings are only normally used as a last resort for medication identification
- There is a possibility to use the score line to help aid handling however this requires further research
- Small tablets are not necessarily easier to swallow and we need to also assess mouthfeel alongside dimensions
- Coating and taste not only help swallowability but are important for identification and memorability
- There are differences in priorities depending on factors such as the setting and the presence of a carer (summarised by Figure 3)
In order to highlight these results more prominently, the authors have updated the Main Findings section as follows:
“This study found that the formulation of oral solid dosage forms has an impact on an older person’s ability to identify, handle and swallow oral solid dosage forms. Figure 2 illustrates the relationship between key characteristics and each stage of the medication taking process. The characteristics can be classified into three main categories: dimensions, appearance and palatability [12].
The dimensions have an impact on all stages; in general, small round tablets (≤ 7 mm) are least accepted amongst older people and their carers and were perceived to have a negative impact on all stages. Interestingly, this includes swallowability due to the need to sense the tablet in the mouth; there is therefore a need to consider mouthfeel alongside the dimensions to optimise this stage of the medication taking process. Preparations with two colours are especially easy to identify and are more memorable as are those with a more distinctive shape. Markings are generally used as a last resort to aid medication identification, and the presence of a score line can further aid handling however this requires further investigation. Palatability includes aspects such as the coating and taste, and while previous studies have identified the importance of this on improving swallowability [Hofmanova], the current study further highlights the importance of these characteristics to aid medication identification and memorability. Several other factors also determine preferences for formulation characteristics, as summarised in Figure 3, and there is a need to ensure these are also considered when designing an older person’s patient centric drug product.” (Pg. 9 Lines 442-450)
8) Which new insight was provided by the carers and health care practitioners? (a request inspired by the paper´s introduction: However, a recent systematic review published in this area found only a single study that involved the views of General Practitioners and no studies that involved formal (paid) or informal (family) carers [12].)
Response 8:
The study highlighted the importance of the role of healthcare professionals in ensuring that an appropriate formulation is prescribed and dispensed. For example, GPs highlighted the need to prescribe tablets rather than capsules for Religious beliefs, social carers highlighted the need to ask for alternative brands of a drug product due to the potential for non-adherence and pharmacists highlighted the potential to assess the patient’s ability to handle medication prior to dispensing (all highlighted within the results section). In order to draw attention to this, the following line has been added to the conclusion:
“In order to ensure the most appropriate formulation is dispensed, health and social care professionals have a key role in assessing the suitability of a formulation…” (Pg. 13 lines 621-624)
Informal carers often have the responsibility of managing multiple medications, and can find this role challenging when characteristics are not optimised. The results throughout have referred to preferences for characteristics of oral solid dosage forms from the point of view of informal carers (for example the need to weigh up the importance of the medication) and these preferences have not previously been explored. The results also highlight the need for a more collaborative approach with healthcare professionals; the following statement has been added to the discussion:
In order to ensure they receive the most appropriate formulation and minimise caregiver burden, there is a need for collaborative partnerships between informal carers and healthcare professionals (21) and this must include exploring issues in relation to characteristics of the dosage form.” (Pg. 11 lines 527-530)
This has also been highlighted in the conclusion:
“In order to ensure the most appropriate formulation is dispensed, health and social care professionals have a key role in assessing the suitability of a formulation and collaborating with both older people and informal carers responsible for the medication administration process.” (Pg. 13 Lines 621-624)
9) How can industry use the results already now, and which issues need to be further scrutinized? For example, what is the best way to use colour? (The results provided many suggestions, so a prioritization is maybe required.)
Response 9:
The conclusion has been updated to provide specific recommendations for the pharmaceutical industry in relation to e.g. size, colour and coating. The updated conclusion provides the following recommendations to ensure optimum acceptance and adherence in this population:
“While there is no “one size fits all” approach, the results highlight a trend towards some key characteristics to optimise these stages of the medication taking process. Smaller tablets (≤ 7 mm) should generally be avoided and where necessary, other characteristics such as colour and mouthfeel should be optimised to aid identification and swallowability. The results further highlight a preference towards brightly coloured and two-colour preparations and the potential to categorise the appearance of medication based on therapeutic indications or the time needed to be taken. This is especially important for older people self-managing multiple medications. Tablets should be easy to handle and the use of a score line or other innovative shapes that tackle dexterity issues in this population requires further investigation. The use of a coating is preferred amongst older people and can help to optimise both medication identification and swallowability.” (Pg. 12 Lines 608-621)
The authors have aimed to explore the issues that require further research under the future work section of the discussion, e.g. the potential for the pharmaceutical industry to explore further the application of 3D printing for this population.
10) Suggestions on how to handle the issue of different brands and generics?
Response 10:
Unfortunately, the issue of different brands and generics is a complex one, and from the results of the present study, one of the most important means of handling this is dependent on the role of health and social care professionals. There is a need for GPs and pharmacists to provide advice when dispensing brands of medication that may appear different in order to ensure older people are aware of a change. However, the results highlight that this role requires further development. This has been highlighted in the discussion on pg. 11, lines 509-512: “Advice from GPs and pharmacists is key to improve the use of these tablets [24]; however, the present study highlights that changes are often not explained to patients, which can result in increased anxiety and reduced acceptance.”
11) It would also be nice if the authors could help the readers with identifying critical medical problems from those that seem manageable or even mostly related to personal taste.
For example:
Line 470 – 472: “Informal carers often find the role of medication management challenging [26] and the present study highlights some difficult decisions they need to make as part of the medication administration process, such as asking for alternative brands due to the potential for non-adherence.” For the reader, the existence of alternative brand may seem like a reasonable solution.
Response 11:
While the existence of an alternative brand may seem like a reasonable solution, the need to balance the medication administration process alongside the needs of the patient and the additional need to ensure the correct brand is dispensed can all increase the pressure on informal carers and add to the caregiver burden. The carer in this case specifically pointed to the need to weigh up the importance of the medication and only ask for an alternative brand if “more important” medications were prescribed as this was an additional burden for her. In addition to the availability of alternative brands, there is therefore also a need for increased support from health and social care professionals. In order to highlight these issues, the following text has been added to the above lines:
“Informal carers often find the role of medication management challenging [26] and the present study highlights some difficult decisions they need to make as part of the medication administration process, such as asking for alternative brands due to the potential for non-adherence. In order to ensure they receive the most appropriate formulation and minimise caregiver burden, there is a need for collaborative partnerships between informal carers and healthcare professionals (21) and this must include exploring issues in relation to characteristics of the dosage form.” (Pg. 11 Lines 527-530)
12) Line 348-350: However, there was also a need to consider the visual aesthetics of the coating, especially for patients with ill mental-health; one patient in a nursing home (caring for those with mental illness) refused ibuprofen with a bright pink sugar coating due to a dislike of brighter colours. Is this regarded as a medical problem, keeping in mind that ibuprofen is available as different types of tablets (brands) and also other formulations?
Response 12:
Again, while other preparations are available, the care professional in this case was highlighting the impact that coating can have on adherence and acceptance. The intentional non-adherence to some coated coloured formulations results in additional caregiver burden for the care professional as they need to ensure the correct brand is ordered in from the pharmacy. Furthermore, the quote is particularly relevant to signify the importance of considering coating alongside the visual aesthetic of the formulation as stated on line 390-391. To highlight this further, the following statement has been added to the above lines:
“However, there was also a need to consider the visual aesthetics of the coating, especially for patients with ill mental-health; one patient in a nursing home (caring for those with mental illness) refused ibuprofen with a bright pink sugar coating due to a dislike of brighter colours. This led to the additional need to ensure the correct brand of ibuprofen was ordered in from the pharmacy. Care professionals suggested white may be a more appropriate colour for these patients, highlighting the importance of considering coating alongside visual identification (Theme 1).” (Pg. 8 Lines 388-389)
13) Line 310-314: Larger tablets may also be difficult to swallow. Preparations that were particularly difficult included calcium and vitamin D preparations, metformin, co-codamol, paracetamol and antibiotics (Appendix A. Table A2). Paracetamol is available as different types of tablets (brands) and also other formulations. Calcium and vitamin D preparations are mainly chewing tablets.
Response 13:
While the authors note and agree the availability of different brands of paracetamol, the majority of participants highlighted the difficulties taking paracetamol and this has been highlighted in Table A2. 6 older people, 2 informal carers, and 5 healthcare professionals all referred to difficulties with paracetamol and this may be due to the general tendency to dispense the cheapest brand of paracetamol available. The authors felt that the high number of participants referring to difficulties with this widely available drug would therefore be important to highlight. While calcium and vitamin d preparations are mainly chewable formulations, the taste of these preparations can be an additional barrier towards optimum adherence and acceptance. For example, quote from HCP8 in table A2. Furthermore, some patients did refer to taking standard calcium and vitamin D tablets, e.g. quote from P8 in table A2. The authors therefore felt it was important to provide a representation of the results and provide real life data of the preparations that participants found particularly difficult.
14) Line 313 – 314: Management techniques deployed by patients included modifying tablets by breaking, re-positioning the head when swallowing, drinking more water and sometimes “ignoring it” because “it’s going to go down eventually.” & line 365 -366: “And they’re not coated so they’re chalky you know, you need a good swallow of water to be able to get them down” (P7). To the reader, this may seem like manageable strategies mainly resolving the issues.
Response 14:
These management techniques deployed by patients did help to eventually ensure they were able to adhere to the medication. However, the need to modify preparations reduces the acceptability of these preparations. Patient acceptability has been defined as “the ability and willingness of a patient to self-administer, and also of any of their lay or professional caregivers, to administer a medicinal product as intended (Pg. 1 Lines 40-41). Patient acceptability is likely to have a significant impact on patient adherence, which can e.g. have an impact on the patient and caregiver quality of life. The need to modify these preparations had an impact on patients’ willingness to take them and there is a need to therefore ensure that preparations are as easy as possible for older people to take or their carers to administer. To make it clear that these strategies impact patient acceptance, the following text has been added:
“Management techniques deployed by patients included modifying tablets by breaking, re-positioning the head when swallowing, drinking more water and sometimes “ignoring it” because “it’s going to go down eventually. The need to deploy management techniques led to reduced acceptability of these preparations.” (Pg. 7 Lines 350-351).
15) Line 371 – 373: Furthermore, capsule formulations containing gelatine require careful consideration; healthcare professionals highlighted that patients often refuse these preparations due to Religious beliefs, leading to intentional non-adherence. Is this a medical problem? (Particularly keeping in mind that most capsules are both halal and kosher.)
Response 15:
This part of the results was taken directly from a quote from a GP, who stated:
“Some of our Muslim patients will refuse capsules on the ground that they feel it’s not Halal, that’s, and that happens even if we, you know, look in some of your product characteristics and say, you know, it’s, you know, a vegetable gelatine or a beef gelatine not pork, you know, even then they’ll say well I’m not sure and I have a feeling, you know, we would make an effort to prescribe tablets rather than capsules for Muslim patients.” (HCP1)
This can lead to a medical problem as although preparations may be halal, the GP suggests patients who are not sure still refuse these formulations, and this intentional non-adherence can therefore be a particular problem. There is therefore a need to specifically change the formulation for some of these patients.
16) Line 477 – 480: Patients use management techniques to overcome difficulties, including drinking more water, splitting/crushing tablet or mixing it with food [28]. Similar management techniques were reported in this study, with participants using characteristics such as the break mark to modify dosage forms; however, this can affect the stability, safety and efficacy of the drug [29]. For this reviewer, it is surprising that the practice of using break marks to break tablets is problematic. Maybe the authors can point to a specific preparation where the stability, safety and efficacy was compromised, encountered during their study?
Response 16:
The authors note the wording of this statement needs improving- the stability, safety and efficacy is more so affected when crushing tablets and opening capsules (29), rather that when using a break mark to break tablets. The above statement has been re-written to reflect this as follows:
“Patients use management techniques to overcome difficulties, including drinking more water, splitting/crushing tablet or mixing it with food [32]. Similar management techniques were reported in this study; however, this can affect the stability, safety and efficacy of the drug” (Pg. 11 Lines 536-537)
17) This is neither to criticize the result nor the method per se, but a call to discuss the results more thoroughly. For example: keeping in mind the wide range of brands and alternatives for most APIs, and the various coping strategies suggested by patients and carers, are there some particularly critical unresolved medical issue/API for which the formulation should be urgently improved, that the authors would like to raise awareness about?
The results provide some weight to the regulatory recommendations provided by both the EMA and FDA with real world data and further identifies the factors that influence patient centric medicine design for older people. For example, the EMA state in their reflection paper that “Patients commonly recognise oral preparations by their size, shape, colour, embossing, rather than by reading the product label” and the current paper highlights the importance of these characteristics on identification and memorability. The paper further supports the recommendations in the EMA reflection paper that older people may be “better classified according to their specific needs rather than chronological age” and goes further to categorise these needs four key areas, as summarised in Figure 3. In addition to highlighting these four key areas, the paper also makes some recommendations to the pharmaceutical industry as to how characteristics can be optimised for this population- please see response 9 above.
18) This reviewer thinks that alternative products/brands and formulations (like oral solutions, orodispersible tablets, transdermal patches, suppositories) have the potential to improve the drug management process for the elderly. The authors, however, point to another solution (line 532-534): Previous studies both within paediatrics and the general adult population have explored the use of 3D printing to improve acceptance [25,37]; further work should explore the use of this technology in the older population, taking into account the recommendations made in this study. It would be nice if the authors could elaborate on why they think 3D printing is the best solution.
Response 18:
The authors agree with the potential for alternative dosage forms to improve the acceptability of medicines, so have added the following sentence:
“In addition, whilst the current study focused on traditional oral solid dosage forms, further work is needed to explore the use of alternative dosage forms, such as orodispersible tablets, to improve patient acceptability,” (Pg. 12 Lines 557-559)
Furthermore, one of the key reasons the authors referred to the particular potential of 3D printing was the importance of an individualised approach towards patient centric medicine design for older people. The heterogeneous nature of this population means several characteristics must be addressed (Figure 3) to provide a truly patient centric product and 3D printing provides an opportunity to address these. The authors have elaborated on the above by including the following sentence:
Previous studies both within paediatrics and the general adult population have explored the use of 3D printing to improve acceptance [29,39]. 3D printing is a particularly valuable tool when designing patient centric drug products for older people as an individualised approach is necessary to take into account the heterogeneous nature of this population. Further work should explore the use of this technology in the older population, taking into account the recommendations made in this study. (Pg. 12 Lines 594-596)
19) Figure 2 is nice (however, as pointed out in the result section, colour can also affect swallowability), but Figure 3 provides information of such a wide and general character, that it can probably be omitted.
Response 19:
Figure 2 highlights the complex relationship between key characteristics and the medication taking process. From the results, the authors feel colour is more suited to the “Medication identification and memorability” category as the results highlighted a refusal to take medication based on a different colour which led to patients being unable to identify the medication. Figure 3, while providing more generic information, is important to highlight the complexities of designing a patient centric drug product for older people. All of the four key categories and the examples provided stem from real life data referred to by participants and the authors therefore feel this Figure is important to reflect the key issues that need to be addressed. Figure 3 is also key to informing future, more targeted research in this area; for example, the need to focus on differing stages of the medication taking process based on the setting.
20) Conclusions: The conclusion is very broad without offering any specific findings nor suggestions, only generalizations. This reviewer has read the paper carefully several times, and seem to have identifies the following:
- Small round tablets (≤ 7 mm) are least accepted amongst older people
- Coating is preferred
- Two-colour preparations are preferred
- Healthcare professionals suggests to assess the patient’s ability to handle medication prior to dispensing
- Healthcare professionals uses score lines for grip
Response 20:
The authors note the need for specific findings in the conclusion, and have updated it as follows:
“Medication adherence in older people is challenging and a key determinant of this is acceptability. Characteristics of oral solid dosage forms are key for ensuring acceptability and, therefore, adherence. There is a need to consider the medication taking process as a whole when optimising these characteristics. Tablets must be visually appealing and memorable, be easy to handle and have optimum swallowability (by considering the dimensions and palatability side by side).
While there is no “one size fits all” approach, the results highlight a trend towards some key characteristics to optimise these stages of the medication taking process. Smaller tablets (≤ 7 mm) should generally be avoided and where necessary, other characteristics such as colour and mouthfeel should be optimised to aid identification and swallowability. The results further highlight a preference towards brightly coloured and two-colour preparations and the potential to use the appearance of medication to categorise tablets based on their therapeutic indications or the time needed to be taken. This is especially important for older people self-managing multiple medications. Tablets should be easy to handle and the use of a score line or other innovative shapes that tackle dexterity issues in this population requires further investigation. The use of a coating is preferred amongst older people and can help to optimise both medication identification and swallowability. In order to ensure the most appropriate formulation is dispensed, health and social care professionals have a key role in assessing the suitability of a formulation and collaborating with both older people and informal carers responsible for the medication administration process.
Environmental, patient, medication and disease characteristics may further lead to a greater emphasis being placed on certain stages of the medication taking process, and these factors therefore also determine preferences for formulation characteristics. Overall, developing an age appropriate dosage design for older people requires a holistic, patient centric approach to improve acceptance and adherence.” (Pg. 12-13 Lines 608-624)
21) Appendix A, Table A2: The presentation of specific preparations is a positive (however, the reviewer´s thoughts on paracetamol and calcium/vitamin D preparations have already been presented). However, the authors should consider whether Table A2 could be shortened a bit.
Response 21:
Table A2 has been shortened as much as possible.

Round 2
Reviewer 2 Report
The manuscript
Author Response
Thank you for your valuable comments.
Reviewer 3 Report
The authors have considered the comments from the reviewer, and answered thoroughly. The method and results are acceptable. However, I still think that the discussion can be improved a bit. My main objection is that the discussion doesn’t distinguish issues that are rooted in the available formulations, from issues mainly rooted in personal beliefs/preferences and organization of the health care system as a whole.
I encourage the authors to consider the following:
Response 2: this comment perhaps caused a misunderstanding: I wanted the authors to elaborate on: These alterations can further complicate treatment by potentially changing the bioavailability, toxicity and stability of medicines [9]. Which medicines? Examples?
The authors write in their response letter: The results throughout have referred to preferences for characteristics of oral solid dosage forms from the point of view of informal carers (for example the need to weigh up the importance of the medication) and these preferences have not previously been explored. This information was not easily extracted from the actual paper.
The authors write in their response letter: 6 older people, 2 informal carers, and 5 healthcare professionals all referred to difficulties with paracetamol and this may be due to the general tendency to dispense the cheapest brand of paracetamol available. I agree, and think this is relevant for the discussion in the paper (i.e.: is the “problem” available suitable formulations, or prize?)
“Patients use management techniques to overcome difficulties, including drinking more water, splitting/crushing tablet or mixing it with food [32]. Similar management techniques were reported in this study; however, this can affect the stability, safety and efficacy of the drug” (Pg. 11 Lines 536-537) I still encourage the authors to provide specific examples: for which medication was the stability, safety and efficacy affected?
When it comes to 3D printing being the solution, I just disagree: I think that is totally unrealistic. But we don’t need to agree on this point. However, I would encourage the authors change their text to:
We believe 3D printing can be a particularly valuable tool when designing patient centric drug products for older people as an individualised approach is necessary to take into account the heterogeneous nature of this population.
Author Response
Response to Reviewer 3 Comments
The authors would like to thank the reviewer for their comments. Please see responses to comments in blue below - specific changes that have been made to the document have been highlighted in bold to aid review. (These have also been made as tracked changes on the article itself).
I encourage the authors to consider the following:
Response 2: this comment perhaps caused a misunderstanding: I wanted the authors to elaborate on: These alterations can further complicate treatment by potentially changing the bioavailability, toxicity and stability of medicines [9]. Which medicines? Examples?
Response 1:
Whilst perhaps slightly outside the scope of the current study, the authors have added some general formulation types where this may be particularly problematic (i.e. modified release preparations and fixed dose combination), as well as some specific examples prevalent for the older population.
The text has been amended, as follows:
“In particular, alterations can further complicate treatment by potentially changing the bioavailability, toxicity and stability of medicines, especially for those oral solid dosage forms with a functional coating and/or modified release properties (such as modified release preparations of co-careldopa, nifedipine and metformin) …” (page 2, lines 64-65)
Indeed, as the cited articles suggest, any modification of a traditional oral solid dosage form can have implications for the physical integrity of the tablet, which can impact on mechanical stability, dose and uniformity of dose (particularly if part doses are given), disintegration and dissolution profiles (as a consequence of an exposed tablet face).
The authors write in their response letter: The results throughout have referred to preferences for characteristics of oral solid dosage forms from the point of view of informal carers (for example the need to weigh up the importance of the medication) and these preferences have not previously been explored. This information was not easily extracted from the actual paper.
Response 2:
The authors consider that this information is presented throughout the manuscript, with some embedded quotes within the main text and several others in the Appendix (Table A2); specifically in section 3.1.2. (Page 5 lines 193-209), section 3.2.1. (Page 6 lines 250-257) and section 3.2.2. (Page 7, lines 288-293), but also several other sections throughout (for example, Page 4 lines 167-169, Page 5 lines 185-186, Page 5 lines 223-225, Page 6 lines 272-274, Page 8 lines 346-3489, Page 8 lines 379-381, Page 11 lines 493-499).
The authors write in their response letter: 6 older people, 2 informal carers, and 5 healthcare professionals all referred to difficulties with paracetamol and this may be due to the general tendency to dispense the cheapest brand of paracetamol available. I agree, and think this is relevant for the discussion in the paper (i.e.: is the “problem” available suitable formulations, or prize?)
Response 3:
The authors agree that this is an important point, and have underlined this further in the discussion with the following text:
“However, this does raise a further issue concerning cost, since oval paracetamol tablets may be more expensive than round paracetamol tablets, whereby community pharmacies may not be remunerated for dispensing the more expensive medication; this further underlines the need for a collaborative approach between patients, carers, healthcare professionals and healthcare providers [9].” (Page 11, lines 511-513)
“Patients use management techniques to overcome difficulties, including drinking more water, splitting/crushing tablet or mixing it with food [32]. Similar management techniques were reported in this study; however, this can affect the stability, safety and efficacy of the drug” (Pg. 11 Lines 536-537) I still encourage the authors to provide specific examples: for which medication was the stability, safety and efficacy affected?
Response 4:
As above, the authors understand the comment and refer back to the previous additions to the text in response 2. Also, the following general comment has been added:
“Similar management techniques were reported in this study; however, this can affect the stability, safety and efficacy of the drug, especially for modified release preparations [33].” (Page 11, lines 505-507)
When it comes to 3D printing being the solution, I just disagree: I think that is totally unrealistic. But we don’t need to agree on this point. However, I would encourage the authors change their text to:
We believe 3D printing can be a particularly valuable tool when designing patient centric drug products for older people as an individualised approach is necessary to take into account the heterogeneous nature of this population.
Response 5:
The authors understand the misgivings of the reviewer around this topic; indeed, 3D printing is currently not a realistic solution from an industrial perspective, although rapid advances in the field do suggest that this could well be a solution, particularly on the smaller scale, in the not too distant future.
Nevertheless, the wording has been changed to the following:
“The authors believe that 3D printing has the potential to be a particularly valuable tool when designing patient centric drug products for older people, as an individualised approach is necessary to take into account the heterogeneous nature of this population” (Page 12, lines 567-570)

This manuscript is a resubmission of an earlier submission. The following is a list of the peer review reports and author responses from that submission.